# Rate of Complications after Hip Fractures Caused by Prolonged Time-to-Surgery Depends on the Patient’s Individual Type of Fracture and Its Treatment

**DOI:** 10.3390/jpm13101470

**Published:** 2023-10-08

**Authors:** Alina Daginnus, Jan Schmitt, Jan Adriaan Graw, Christian Soost, Rene Burchard

**Affiliations:** 1Faculty of Medicine, University of Marburg, 35037 Marburg, Germany; 2Department of Orthopaedics and Traumatology, University Hospital of Giessen and Marburg, 35043 Marburg, Germany; 3Department of Orthopaedics and Trauma Surgery, Lahn-Dill-Kliniken, 35683 Dillenburg, Germany; 4Department of Anesthesiology and Intensive Care Medicine, Ulm University Hospital, 89070 Ulm, Germany; 5Institute for Empirics & Statistics, FOM University of Applied Sciences, 45141 Essen, Germany

**Keywords:** hip fracture, time to surgery, complications, intracapsular hip fracture, extracapsular hip fracture

## Abstract

Introduction: Hip fractures are common injuries in the elderly and are usually treated with timely surgery. While severe postoperative complications are reported for up to 10% of patients, many studies identified predictive factors for the occurrence of complications postoperatively. A controversially discussed factor is “time-to-surgery”. The aim of the study was to examine if time-to-surgery was associated with the occurrence of complications and if the complication rate differed between the patient individual fracture types of intracapsular on the one hand and extracapsular hip fractures on the other hand. We hypothesized that time-to-surgery had less impact on complications in intracapsular hip fractures compared to extracapsular ones, and therefore, guidelines should pay attention to the patient individual case scenario. Materials and Methods: All patients who were admitted to the Department of Trauma and Orthopaedic Surgery of an academic teaching hospital for hip fracture surgery (*n* = 650) over a five-year period were included in the study. After the application of the exclusion criteria, such as periprosthetic or pathologic fractures, cases needed immediate surgical treatment, and after outlier adjustment, 629 cases remained in the study. Hip fractures were classified into intracapsular fractures (treated by hip arthroplasty) and extracapsular fractures (treated by intramedullary nailing osteosynthesis). The occurrence of severe complications in patients treated within 24 h was compared with patients treated later than 24 h after injury. For statistical evaluation, a multivariate logistic regression analysis was performed to investigate the impact of time-to-surgery interval on the occurrence of complications. Results: Patients with an extracapsular fracture, which was treated with intramedullary nailing (44.5%), rarely suffered a serious complication when surgery was performed within 24 h after injury. However, when the interval of the time-to-surgery was longer than 24 h, the complication rate increased significantly (8.63% vs. 25.0%, *p* = 0.002). In contrast to this finding in patients with intracapsular fractures (55.5%), which were treated with cemented arthroplasty, complication rates did not depend on the 24 h interval (26.17% vs. 20.83%, *p* = 0.567). Conclusions: The occurrence of complications after surgical treatment of hip fractures is associated with the time interval between injury and surgery. A 24 h time interval between injury and surgical procedure seems to play a major role only in extracapsular fractures treated with osteosynthesis but not in intracapsular fractures treated with arthroplasty. Therefore, guidelines should take notice of the patient individual case scenario and, in particular, the individual hip fracture type.

## 1. Introduction

Hip fractures are one of the most common bone injuries of the musculoskeletal system in the elderly, and the incidence per year rises with age [1,2]. Across Europe, incidences are 0.5–1.6% per year [3,4]. For example, in Germany, annual incidences for those over 65 years of age range from 0.6 to 0.9% [5,6,7]. In industrialized countries, there is an expected yearly increase in the incidence of hip fractures of 3–5%, and therefore, a doubling of cases can be anticipated by 2040 [8,9]. Hip fractures predominantly occur in women, with men having only half the estimated risk [10]. For people in the age group above 65 years, hip fractures are the most common cause of admission and hospitalization to an orthopedic or trauma department. The prognosis after these injuries is unfavorable since a worldwide 1-year mortality rate of 22% is calculated based on a review article including 229,851 patients [11]. Hip fractures are a worldwide major public health burden with high costs for health systems and social care through hospital stays and subsequent rehabilitation or home care. Costs for fractures close to the hip add up to two to four billion EUR per year [1,12,13,14].

Hip fractures are usually treated with surgery depending on the fracture type, and a timely operation of these serious injuries results in good outcomes and fewer complication rates [15]. Potential complications such as wound infections or embolisms can lead to dependency, immobility, impaired quality of life, and death. [7,16]. For instance, Seong and colleagues and Klestil and colleagues described a one-year mortality rate of 14–36% after surgery [2,17]. These results are in line with data described in a review by Downey and colleagues [11]. Furthermore, there is a significant rate of postoperative complications in both, systemic complications such as myocardial infarction, deep vein thrombosis, pulmonary embolism, and urinary tract infection and surgical complications such as wound infections, expanded hematomas, and any other serious condition with the need for revision surgery [1,2]. In addition to others, Palma and colleagues described a rate for severe complications ranging from loss of mobility or loss of independence up to death between 8 and 28% [18,19,20].

Many studies revealed predictive factors for the occurrence of perioperative complications after hip fracture, such as gender, age, comorbidities, systemic anticoagulation, and the general physical status at the time of the injury [1,11,17,19,21,22,23,24,25,26,27,28,29,30,31,32]. The most controversial factor recognized and examined in multiple studies on the therapy of hip fractures is the “time-to-surgery” interval [19,33,34]. While some studies found an increased complication rate for surgical procedures performed after 24 h, other study groups found disadvantages for a too-early operation or detected no significant association between the time-to-surgery interval and complication rates [19,21,22,23,24,25,26,35]. Some guidelines, such as the “German guideline for the treatment of femoral fracture close to the hip joint”, recommend surgery for these fractures without differentiating the fracture type patient individual within 24 h after injury [27]. Others recommend treatment within 48 h after hospital admission [36,37]. However, there is no consensus about the optimal time window for treatment in international literature.

However, none of the aforementioned studies to date examined fracture type or type of surgical procedure in terms of the optimal time to surgery. Due to the completely different pathoanatomical conditions of the various fracture types, the distinction between intra- and extracapsular fractures should be of particular interest [38]. Analyzing Swedish registry data, Mattisson and colleagues recommended surgery within 36 h for extracapsular fractures due to the more problematic results [39]. However, they could not give any recommendations for intracapsular fractures. In addition, the common surgical approach to address these two different fracture types differs too: while intracapsular fractures are usually treated with arthroplasty, extracapsular fractures are mainly treated with osteosynthesis, e.g., with intramedullary nailing [40].

Therefore, the aim of this study was to investigate whether the occurrence of complications after treatment of hip fractures differs according to the patient individual type of fractures and the specific treatment procedures.

## 2. Materials and Methods

### 2.1. Ethical Approval

The ethics committee of the State Medical Chamber of Westfalia-Lippe approved the present study according to the ethical standards (number of ethical approval: 2015-497-f-S). Written informed consent of included patients was waived by the ethics committee because of the retrospective nature of the presented study. All methods were performed in accordance with the relevant guidelines and regulations as stated in the Helsinki Declarations.

### 2.2. Study Design

This single-center retrospective observational study includes a reanalyzed subset of data from a patient population from a previous study examining the gender-specific circumstances of required transfusion in the setting of hip fractures [41]. All patients admitted to the Department of Trauma and Orthopaedic Surgery of a 595-bed-academic teaching hospital of the University of Marburg for hip fracture surgery (*n* = 650) in the years 2010 to 2014 were included in the study. The following exclusion criteria were stated: periprosthetic fracture, fractures that had to be treated directly by osteosynthesis such as intracapsular fractures of young and healthy patients (age < 65), pathologic fractures such as metastasis, postoperative conditions that forbid an immediate full weight-bearing, and fractures including the pelvic bone. After application of the exclusion criteria and after outlier adjustment, 629 cases remained in the study collective. According to the major study target, all hip fractures were classified into intracapsular fractures (treated by hip arthroplasty) and extracapsular fractures (treated by intramedullary nailing).

### 2.3. Surgical Technique and Postoperative Care

All surgical procedures in both study groups were performed under general anesthesia. Neither during osteosynthesis of extracapsular fractures nor during arthroplasty of intracapsular fractures, tranexamic acid was used. Disinfection administration and perioperative care, such as warming, pain and physiotherapy, and patient blood management, were performed according to the national guidelines and the recommendations of the World Health Organization (WHO).

#### 2.3.1. Intracapsular Fractures

Since urgent cases of intracapsular hip fractures, just as younger patients needed immediate osteosyntheses with, for example, cannulated screws, were excluded, all intracapsular fractures of the presented study collective were treated by hip arthroplasty. The surgical procedure was performed under single-shot antibiotic prophylaxis 30–45 min prior to surgery, applying 2 g of Cefazoline intravenous. A muscle-protecting anterolateral approach using the so-called Watson–Jones interval was performed, and after resection of the fractured femoral neck by a saw cut, the femoral head was removed, too. Femoral bone was prepared for the implantation of a cemented straight stem. In all cases, the Excia T Standard stem was implanted (Aesculap AG, Tuttlingen, Germany). After cementing a bipolar steel head according to the patient, the individual diameter of the removed bony head was applied (Bipolar Head, Aesculap AG, Tuttlingen, Germany). After the insertion of two suction drains into the subfascial and subcutaneous tissue, the wound was closed and draped. Postoperative treatment included immediate gait training with full weight-bearing on crutches beginning from day 1 after surgery. Prophylaxis of embolic complications was performed by standardized anticoagulation by low molecular weight heparin.

#### 2.3.2. Extracapsular Fractures

Extracapsular fractures were treated by intramedullary nailing osteosynthesis. The surgical procedure was performed under single-shot antibiotic prophylaxis 30–45 min prior to surgery applying 2 g of Cefazoline intravenous. First fracture reduction was established using a classic reduction table. Therefore, the injured leg was fixed and repositioned under X-ray control in anterior posterior view and in lateral view. To fix the reposition, an intramedullary nail was inserted (Targon PFT, Aesculap AG, Tuttlingen, Germany). This proximal femur nail has a gliding femoral neck screw and an additional anti-rotation pin above the neck screw. Distally the nail was fixed with one bicortical screw. Wound closure included a multilayer closing and a sterile draping after the application of one subfascial suction drain. Postoperative treatment included immediate gait training with full weight-bearing on crutches beginning from day 1 after surgery. Prophylaxis of embolic complications was performed by standardized anticoagulation by low molecular weight heparin.

### 2.4. Data Collection

Cases were identified by ICD (International Statistical Classification of Diseases and Related Health Problems) codes (S72.00-08 and S72.10-11) using the hospital patient data management system (MCC Meierhofer^®^, Meierhofer AG, Munich, Germany). The following variables were extracted from the electronic patient data files: demographic patient data such as age, sex, date of accident, time to surgery interval, length of surgery (incision to end of suture), patient individual type of fracture, surgical treatment such as arthroplasty or osteosynthesis, medication profile, overall length of hospital stay, admittance to intensive care unit, cumulated costs of the hospital stay, patients’ residential status (living at home or living in a nursing home), and common laboratory values (hemoglobin concentration, International Normalized Ratio (INR), blood glucose, electrolytes such as natrium, kalium, calcium, and kidney retention parameters). Presence of a comorbidity was considered if the following previous illnesses were identified: dementia, symptomatic heart failure, previous ischemic stroke, gait disorder, living in a nursing home before hospitalization, and the presence of polypharmacy, which was defined as 4 or more medications taken.

### 2.5. Study Outcome Parameters

Occurrence of severe perioperative complications was defined as primary outcome parameter. The need for revision surgery and the occurrence of any deep wound infection were considered severe surgical complications. In addition, occurrence of severe systemic perioperative complications was defined as secondary outcome parameter, including death, thromboembolic events, perioperative pneumonia, and urinary tract infection. Thromboembolic events were defined as pulmonary embolism, deep vein thromboembolism, ischemic stroke, and myocardial infarction.

### 2.6. Statistical Analysis

The statistical analysis of the data was conducted using the R software package, version 4.0.4, developed by the R Foundation for Statistical Computing in Vienna, Austria. Categorical data and bivariate relationships were assessed using two statistical tests: Pearson’s Chi-Square test of independence and the two-sample proportion test. For the bivariate analysis, results are presented in terms of absolute numbers and frequencies (%) or means unless otherwise specified. In addition, the standard errors are also reported. To delve deeper into the relationships between variables and control for potential confounding factors, multivariate analysis was employed, specifically logistic regression. This method enables us to identify the influence of multiple factors simultaneously and allows us to control for potential influence factors and obtain a more accurate understanding of the factors affecting the outcome, reducing the risk of selection bias in our study. Standard errors presented in parentheses alongside the regression results were included. A significance level of *p* < 0.05 was set as the threshold for statistical significance.

## 3. Results

Table 1 shows the characteristics of the 629 patients undergoing surgical hip fracture treatment. There were 449 (71.38%) female and 180 (28.62%) male patients (3:1 female to male ratio). The mean age of all patients was 79.29 years (range 20–102). In addition, a mean length of stay in the hospital of 14.44 days (sd 6.92 days, range 1–79 days) was recognized.

Based on the fracture type, two groups were compared: patients with intracapsular hip fractures (44.52%) and patients with extracapsular fractures (55.48%). Patients with an extracapsular fracture rarely suffered a serious complication if the surgical treatment was performed within the 24 h interval, but when the 24 h window was exceeded, complications increased significantly in this group (27 (8.63%) vs. (9 (25.0%), *p* = 0.002). In the group of intracapsular fractures, the complication frequency did not differ between a surgical treatment within 24 h on the one hand and a treatment minimum 24 h after the injury on the other hand (67 (26.17%) vs. 5 (20.83%), *p* = 0.567).

To reduce the probability of a possible selection bias, multivariate analysis was performed using logistic regression and stepwise integration of patient characteristics of age, sex, place of residence, comorbidities, and hospital length of stay. The results in Table 2 show a significant increase in the probability of complications for the extracapsular fracture type and duration to surgery greater than 24 h compared to the intracapsular fracture type for models 1–3 (*p* < 0.05). The addition of the control variables leads only to marginal changes in the regression coefficients but not to a change in the significant interaction. This allows us to show that the effect is not biased by the sociodemographic characteristics of the study participants. The significant interaction effect of the most comprehensive model 3 (probability to suffer from a complication regarding the time to surgery (within 24 h vs. >24 h)) is shown in Figure 1, allowing a more intuitive interpretation of the findings.

In addition, logistic regression analysis (most comprehensive model 3) provides results for the influence of the control variables and reveals that the probability of occurrence of complications increased with increasing age (b = 0.040, OR = 1.041, *p* < 0.05). The probability of occurrence of complications also increased if dementia (b = 0.999, OR = 2.714, *p* < 0.01) or heart failure (b = 1.260, OR = 3.525, *p* < 0.001) was present.

## 4. Discussion

This retrospective study demonstrates that patients with an extracapsular hip fracture rarely suffered a serious perioperative complication when surgery was performed within 24 h after injury. However, when the interval of the time-to-surgery was >24 h, the complication rate increased significantly. In patients with intracapsular fractures, complication rates did not differ between patients treated within 24 h and patients with surgery performed >24 h after injury.

Due to the demographic development of humankind and medical progress, geriatric medicine is one of the challenges of modern medicine [8]. Hip fractures are among the most common traumas of the elderly, and because these people are usually pre-diseased, a higher risk for the occurrence of perioperative complications is present [16]. The study collective was comparable to other studies, including epidemiologic parameters such as age, gender distribution, BMI, blood values, and presence of comorbidities [2,7,11,15,16,17]. Furthermore, the general complication rate (9.54% of all cases) was comparable to the complication rates reported from similar study cohorts [18,19,20].

### 4.1. The Time to Surgery Interval in Hip Fractures

The optimal time for surgery associated with the lowest risk of complications after hip fracture surgery has been the subject of numerous studies in the literature. While some authors found advantages for surgery within 24 h, others could not confirm these results [1,17,19,21,22,23,24,25,26,28,29,30,31,32,35,42,43,44]. These heterogeneous results thus leave room for interpretation also for national guidelines. Taken together, most of the studies recommend surgery as soon as possible for all types of hip fractures without considering individual fracture types. In the German guidelines, for example, treatment within 24 h is recommended as mandatory [27]. In contrast, U.S. and U.K. guidelines recommend surgical treatment of hip fractures within 48 h of admission [36,37]. The American Academy of Orthopaedic Surgeons (AAOS) recommends surgery within 48 h and claims moderate evidence [17]. The discrepancy between recommendations can be explained by the heterogeneity of studies and the lack of a clearly definable cutoff for the safest time between admission and the start of surgery.

### 4.2. Impact of Patient Individual Hip Fracture Type

In all the studies mentioned, the type of fracture has not been considered differentiated in the analysis of the results [1,17,19,21,22,23,24,25,26,28,29,30,31,32,35,42,43,44]. Reasons for the assumed influence of the fracture type on the complication rate are mainly to be found in the anatomy of the hip joint or the proximal femur [40]. Intracapsular fractures usually have low primary blood loss because of the confinement by the capsule itself [38]. Extracapsular fractures, on the other hand, may have much more extensive bleeding into the soft tissues of the entire femur [38]. Therefore, Harper and colleagues describe a “hidden blood loss” of extracapsular fractures compared to intracapsular fractures [38]. These findings are well explained by the anatomy of the proximal femoral region. Intracapsular fractures usually have very limited blood loss until the time of surgery because the rough and tight capsule limits the blood loss to only a few milliliters. In contrast, blood loss in extracapsular fractures differs substantially due to the absence of the limiting capsule. Bleeding from the medullary cavity of the largest human bone is limited only by the capacity of the soft tissue space. Therefore, significant blood loss up to hemorrhagic shock is known from traumatic extracapsular fractures of the femur [45]. Stacey’s research group was also able to show significant differences in preoperative blood loss related to the patient individual fracture type in hip fractures [46]. In intracapsular hip fractures, hemoglobin levels dropped about an average of 1.1 g/dL. On the other hand, extracapsular fractures showed much more extensive blood loss with an average loss in hemoglobin concentration of 1.7 g/dL. The differences in blood loss were also evident in immediate postoperative blood count measurement as a consequence of the already higher preoperative blood loss. Thus, patients with intracapsular fractures left the operation theatre with an average hemoglobin value of 10.1 g/dL, while those with extracapsular fractures had a value of only 8.8 g/dL [46]. Further examination of the pathophysiological processes in the context of increased blood loss, especially in mono-injury of the large tubular bones, reveals a significantly increased inflammatory response with potential consequences for several organ systems in various animal models in pigs and mice [47,48]. A significantly increased expression of Interleukin-6-related genes, as well as the infiltration of polymorphonuclear (PMN) leukocytes into the tissue, could be detected [47]. In particular, the lungs and liver were affected. Furthermore, increased age was found to be a predictive factor for exacerbating liver inflammation and, for this reason, also an additional disorder of coagulation [48].

Due to the lack of limitation of the peri-trochanteric bone regions by a rough capsule, extracapsular hip fractures are associated with significantly increased soft tissue damage, especially in the skeletal muscles resulting from trauma by the sharp-edged fragments of the dislocated bone parts. Such soft tissue damage leads to both an increased inflammatory response and increased blood loss. In this context, Pierce and Pittet described a vicious circle of soft tissue damage and inflammatory response [49]. The soft tissue damage and the subsequent response result in damage to the endothelium with a corresponding change in permeability and subsequent soft tissue edema. This edema further leads to worsened tissue perfusion and oxygenation. This, in turn, increases the soft tissue damage and closes the vicious circle [50]. Infiltration of tissue, particularly muscle, by inflammation-mediating cells and mediators peaks between 12 and 24 h after trauma [50]. These pathophysiological correlations further suggest that early therapy might be useful in extracapsular hip fractures with concomitant soft tissue trauma.

In summary, extracapsular hip fractures offer a higher risk profile regarding a prolonged time to surgery due to their anatomical and pathophysiological circumstances. These are not found to the same extent in intracapsular fractures, and therefore, the reduction and stabilization of extracapsular fractures might be of increased priority.

In addition, intracapsular fractures have a correspondingly lower influence on the entire body because of the usually better preoperative physical condition of the patients [51]. Hershkovitz and Rutenberg recently reported that the patient populations with extracapsular and intracapsular hip fractures differed in physical condition, with patients with intracapsular fractures having a better physical condition on average [52]. The patient cohort with intracapsular fractures had a slightly higher proportion of male gender, a higher level of education, and more frequent access to home care than those with extracapsular fractures. This is also reflected in the fact that postoperative recovery is faster in intracapsular fractures and is more often associated with a favorable outcome [51,52]. The results found in this study support the differentiated view and suggest that individual consideration of the patient-specific fracture type is useful in hip fractures.

Based on the high frequency of hip fractures in the elderly and, therefore, the high relevance of this disease for healthcare systems particularly in the Western world, guidelines support maintaining the quality of treatment at a consistently high level [27,36]. Nevertheless, as this study supports, patient-specific aspects might also be considered in the decision-making process [40]. Besides the medical quality of treatment, appropriated resource allocation is a growing factor for each healthcare system with an impact on healthcare economics [12,14]. The high costs of holding hip fractures coupled with a growing shortage of specialists challenges many hospitals in the timely treatment of hip fractures [53]. In particular, the treatment of intracapsular fractures of the elderly by means of arthroplasty is in focus here [40]. The fact that these latter cases are now less prone to delayed surgery could be a finding of great interest with a corresponding health policy impact.

### 4.3. Limitations

Although the work presented is based on a large patient population, this study is limited by the retrospective and single-center study design. While the surgeons performing the operations remained constant during the study period, intracapsular fractures of younger patients requiring acute care (<6 h) by primary osteosynthesis were excluded.

In summary, the results currently available in the international literature on the question of the best possible surgical timing in patients with hip fractures do not provide clear evidence for a 24 h rule. The lack of data on differentiation for different patient individual fracture types and the influence of fracture type on the occurrence of complications after hip fracture has not been adequately studied. The present study provides new additional insights that could facilitate the approach in orthopedic and trauma surgery departments through a patient-specific treatment approach to hip fracture surgery and a more differentiated application of the guideline recommendations.

## 5. Conclusions

Occurrence of a severe complication after surgical treatment of hip fractures is associated with the time interval between injury and surgery. A 24 h time interval to surgery seems to play a role only in extracapsular fractures treated with osteosynthesis but not in intracapsular fractures treated with arthroplasty. Therefore, recommendations of current national and international guidelines on hip fracture surgery should be reevaluated with respect to the patient individual hip fracture type.

## Figures and Tables

**Figure 1 jpm-13-01470-f001:**
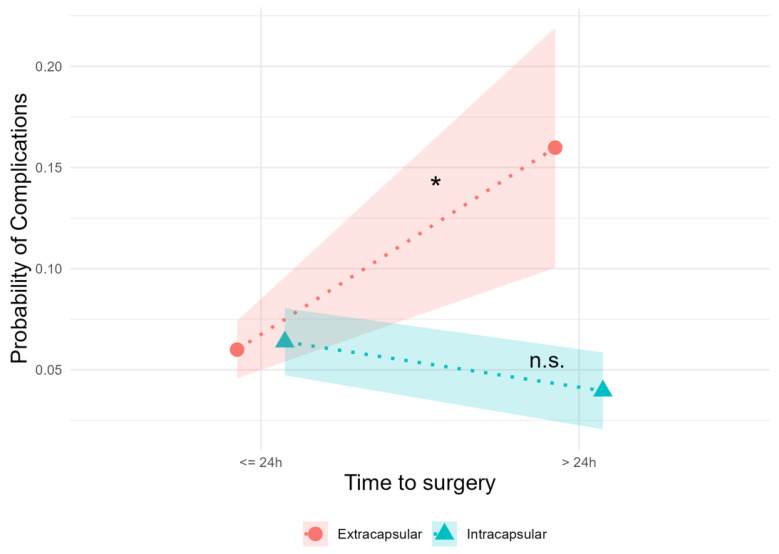
Comparison of the complication rates between the extra- and intracapsular groups. A significant increase in the complication probability can only be seen in the extracapsular group instead of the intracapsular group (* = *p* < 0.05, n.s. = not significant).

**Table 1 jpm-13-01470-t001:** Patient characteristics (*n* = 629).

Age, mean (sd)	79.29 years (11.94 years)
Range	20–102 years
Gender, n (%)	
female	449 (71.38%)
male	180 (28.62%)
Comorbidities, *n* (%)	
dementia	138 (21.94%)
symptomatic heart failure	241 (38.31%)
stroke	88 (14.00%)
gait disorder	145 (23.05%)
Place of residence, *n* (%)	
live at home	507 (80.60%)
live in a nursing home	98 (15.58%)
unknown	24 (3.82%)
Medication, *n* (%)	
none	90 (14.31%)
1–3	145 (23.05%)
4–6	133 (21.14%)
>7	351 (55.80%)
INR, *n* (%)	
≤1.5	566 (91.14%)
>1.5	55 (8.86%)

**Table 2 jpm-13-01470-t002:** Logistic regression model in which predictors of complication occurrence are added stepwise in models 1, 2, and 3.

	Complication		
	(1)	(2)	(3)
Intracapsular fracture	0.063	−0.025	0.067
	(0.314)	(0.317)	(0.330)
Time to surgery > 24 h	1.262 **	1.200 **	1.093 *
	(0.434)	(0.448)	(0.476)
Age		0.057 ***	0.040 *
		(0.017)	(0.019)
Male		−0.142	−0.329
		(0.338)	(0.360)
Residence: live at home			0.429
			(0.436)
Residence: live in a nursing home			0.275
			(0.418)
Residence: unknown			1.146
			(0.646)
Dementia			0.999 **
			(0.323)
Symptomatic heart failure			1.260 ***
			(0.319)
Stroke			0.093
			(0.434)
Gait disorder			−0.026
			(0.371)
Length of stay			0.029
			(0.017)
Intracapsular fracture * Time to surgery > 24 h	−1.559 *	−1.453 *	−1.596 *
	(0.679)	(0.691)	(0.716)
Constant	−2.360 ***	−6.932 ***	−7.259 ***
	(0.201)	(1.439)	(1.708)
Observations	629	629	629
Log Likelihood	−193.936	−185.973	−169.866
Akaike Inf. Crit.	395.872	383.946	367.731

Notes: * Significant at the 5 percent level. ** Significant at the 1 percent level. *** Significant at the 0.1 percent level.

## Data Availability

The datasets used and/or analyzed during the current study are available from the corresponding author upon reasonable request.

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
