# Peer review of "Rate of Complications after Hip Fractures Caused by Prolonged Time-to-Surgery Depends on the Patient’s Individual Type of Fracture and Its Treatment"

_jpm, 2023, doi:10.3390/jpm13101470_

Round 1

Reviewer 1 Report

The research aim is to analyze the correlation between postoperative complications and time to surgery after injury for intra- and extracapsular hip fractures.

The abstract is structured appropriately.  The sentence in line 20 “Retrospective study of patients (n=629) with hip fracture” has no verb and should be revised.

The introduction transposes the research into the topic and formulates the objective of the study at the end.

In the methodology section, the stages of the research are presented. I suggest dividing subsection 2.1 in three different subsections, namely Study design, Data collection and Surgical technique and post-operative care (here please describe the procedures performed as hip arthroplasty and intramedullary nailing is too general).

In the results sections the are some editing errors.  The header of the two Tables should be revised as they are empty. In Figure 1 the meaning of the symbol “*” and “n.s.” should be provided as it is confusing.

The discussions interpret the research results and relate them to other findings from scientific literature. However, more particular examples on the potential negative effects of time-to-surgery in elderly patients with hip fractures should be provided.

The conclusions are concise and clear.

The references are adequate but can be extended as suggested above.

Author Response

Reviewer #1 – revisions are highlighted in yellow:

  1. The abstract is structured appropriately. The sentence in line 20 “Retrospective study of patients (n=629) with hip fracture” has no verb and should be revised.

We thank the reviewer for exposing this mistake. Therefore, we revised the Materials and Methods section of the Abstract (lines 45-48, page 1 of the revised manuscript).

  1. In the methodology section, the stages of the research are presented. I suggest dividing subsection 2.1 in three different subsections, namely Study design, Data collection and Surgical technique and post- operative care (here please describe the procedures performed as hip arthroplasty and intramedullary nailing is too general).

We thank the Reviewer for this valuable suggestion. According to the suggestions we have divided the Methodology section in the recommended subsections. In addition, a subsection “Surgical technique and postoperative care” was added to the main section 2 with a full description of the performed procedures and the treatment after surgery on pages 4-6 of the revised version of the manuscript.

  1. In the results sections the are some editing errors. The header of the two Tables should be revised as they are empty. In Figure 1 the meaning of the symbol “*” and “n.s.” should be provided as it is confusing.

We agree with the Reviewer and have asked the Editorial office to assist with editing of the respective Tables since the mentioned problems appear to be editing errors of the MDPI style. In addition, the Figure legend of Figure 1 was rewritten and the respective abbreviations (* meaning significant p<0.05, n.s. meaning not significant) were described in the legend (page 18 of the revised manuscript).

  1. The discussions interpret the research results and relate them to other findings from scientific literature. However, more particular examples on the potential negative effects of time-to-surgery in elderly patients with hip fractures should be provided.

We thank the reviewer for highlighting this important point. Therefore, we enhance the whole Discussion section including examples on the potential negative effects of time-to-surgery in elderly patients with hip fractures. Therefore, on pages 8-10 of the revised manuscript we added a completely new subsection 4.2.

  1. The references are adequate but can be extended as suggested above.

According to the reviewer’s and editor’s suggestions we revised the references and add some more of relevant and freshly published ones.

Reviewer 2 Report

Thank you for submitting this manuscript

you wrote in the introduction “many studies revealed…………..and this sentence ended with one reference (20). It was expected to find many references

the discussion need to be rewritten because it is written like an introduction with the knowledges from the literatures rather than commenting on the results of this work and your explanation of the results and comparing them with the previous similar work

Average

Author Response

Reviewer #2 – revisions are highlighted in yellow:

  1. you wrote in the introduction “many studies revealed…” and this sentence ended with one reference (20). It was expected to find many references.

Thank you very much for highlighting this point. We have now included a more appropriate number of references to underline the respective statement in the Introduction section on page 3, lines 96-98 of the revised version of the manuscript.

  1. the discussion needs to be rewritten because it is written like an introduction with the knowledges from the literatures rather than commenting on the results of this work and your explanation of the results and comparing them with the previous similar work.

We thank the reviewer for highlighting this important point. Therefore, we enhance the whole Discussion section including examples on the potential negative effects of time-to-surgery in elderly patients with hip fractures. Therefore, on pages 8-10 of the revised manuscript we added a completely new subsection 4.2.